# NEURAL GRAPHICAL MODELLING IN CONTINUOUS-TIME: CONSISTENCY GUARANTEES AND ALGORITHMS

**Alexis Bellot**[*]
Columbia University, USA
ab5305@columbia.edu

**Kim Branson**
GlaxoSmithKlein, USA
kim.m.branson@gsk.com

**Mihaela van der Schaar**
University of Cambridge, UK
The Alan Turing Institute, UK
University of California, Los Angeles, USA
mv472@cam.ac.uk

## ABSTRACT

The discovery of structure from time series data is a key problem in fields of study working with complex systems. Most identifiability results and learning algorithms assume the underlying dynamics to be discrete in time. Comparatively few, in contrast, explicitly define dependencies in infinitesimal intervals of time, independently of the scale of observation and of the regularity of sampling. In this paper, we consider score-based structure learning for the study of dynamical systems. We prove that for vector fields parameterized in a large class of neural networks, least squares optimization with adaptive regularization schemes consistently recovers directed graphs of local independencies in systems of stochastic differential equations. Using this insight, we propose a score-based learning algorithm based on penalized Neural Ordinary Differential Equations (modelling the mean process) that we show to be applicable to the general setting of irregularly-sampled multivariate time series and to outperform the state of the art across a range of dynamical systems.

## 1 INTRODUCTION

This paper deals with learning directed graphs from a combination of temporal data and assumptions on the parameterization of the underlying structural dynamical system. Graphical models can offer a parsimonious, interpretable representation of the dynamics of stochastic processes, and have proven to be especially useful in problems involving complex systems, non-linear associations and chaotic behaviour that are characteristic in a wide array of applications in biology (Trapnell et al., 2014; Qiu et al., 2017; Bracco et al., 2018; Raia, 2008; Qian et al., 2020), neuroscience (Friston et al., 2003; Friston, 2009) and climate science (Runge, 2018; Runge et al., 2019). In these contexts, inferring graphical models from temporal data subject to practical limitations as to how finely and regularly each variable can be measured over time is a longstanding challenge.

Time series data is often assumed to be a sequence of observations from an underlying process evolving continuously in time. This underlying representation is fundamental to define the semantics of dependencies between sequences. Time defines an asymmetry between dependencies in dynamical systems, distinguishing between local, direct dependencies that occur over infinitesimal time intervals not mediated by other variables in the system and indirect dependencies that necessarily occur over longer time frames. In many applications, the underlying structural model is formalized as the state of random variables (e.g. $\mathbf{x}(t) \in \mathbb{R}^d$) contemporaneously influencing the rate of change of the same or other variables (e.g. $d\mathbf{x}(t)$),

$$d\mathbf{x}(t) = \mathbf{f}(\mathbf{x}(t))dt + d\mathbf{w}(t), \qquad \mathbf{x}(0) = \mathbf{x}_0, \qquad t \in [0, T], \qquad (1)$$

---

[*]Work primarily conducted while at the University of Cambridge and at the Alan Turing Institute.

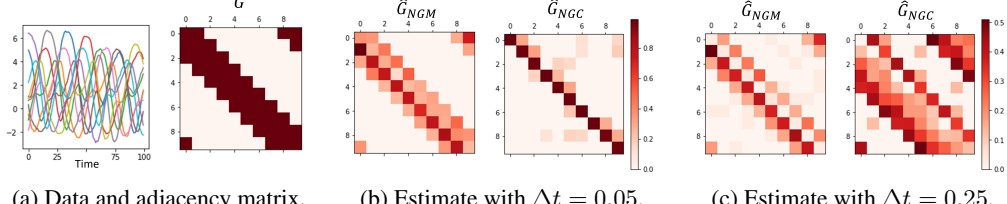

(a) Data and adjacency matrix.    (b) Estimate with $\Delta t = 0.05$.    (c) Estimate with $\Delta t = 0.25$.

Figure 1: Visual comparison of the true and learned adjacency matrices $G$ of a 10-variable Lorenz system (see section 4.1). $\hat{G}_{\text{NGM}}$ (the proposed continuous-time approach) and $\hat{G}_{\text{NGC}}$ (Neural Granger Causality (Tank et al., 2018)) are estimates of continuous and discrete-time algorithms respectively. Panel (a) shows a data sample and the true adjacency matrix, panel (b) shows estimates with higher frequency of observation ($\Delta t = t_i - t_{i-1} = 0.05$) and panel (c) with lower frequency of observation ($\Delta t = 0.25$). The heat scale gives the strength of estimated functional interactions. An explicitly continuous-time model is more accurate and more robust to the sampling frequency than discrete-time alternatives for graphical modelling in dynamical systems.

where $\mathbf{w}(t)$ a $d$-dimensional standard Brownian motion and $\mathbf{x}_0$ is a Gaussian random variable independent of $\mathbf{w}(t)$. The functional dependence structure of the vector field $\mathbf{f}$ defines a directed graph $\mathcal{G}$ and associated adjacency matrix $G \in \{0, 1\}^{d \times d}$, i.e., $G_{ij} = 1$ if and only if $x_j$ appears as an argument of $f_i = [\mathbf{f}]_i$. The problem of structure learning is to search over the space of graphs compatible with the data, but the pattern of observation in dynamical systems emphasize a number of differences with respect to classical graphical modelling with static data or explicitly discrete-time stochastic process.

- Observed data is sampled at a sequence of (often irregular) time points $(t_1, \ldots, t_n)$ and is systematically subsampled. Most work on graphical modelling with time series data assume a fundamentally discrete parameterization of the underlying structural model (e.g. based on vector autoregression models). Associations in discrete-time in general do not correspond to the structure of the underlying dynamical system and are highly dependent on the interval between observations. The same subsampled discrete model may disaggregate to several continuous models, which are observationally equivalent at the subsampled frequency, see e.g. (Runge, 2018; Gong et al., 2015; Danks & Plis, 2013) and a worked example in Appendix A. The realm of problems that involve irregularly-sampled data are fundamentally out of scope in discrete-time in general. We complement this point in Figure 1 with an illustration of our performance results comparing a state of the art discrete-time graphical modelling method with our proposed continuous-time counterpart that is shown to be more accurate and more robust to the frequency and irregularity of sampling.

- Discrete samples are not independent which can (and does) increase the sample complexity. An increasing sample frequency will produce an increasing number of distinct samples. However, samples become more dependent, and intuitively one expects that there is limited information to be harnessed from a given time interval $[0, T]$. Learning performance depends on the number of independent samples which is a function both of the number of samples $n$ and the length of the observed interval $T$.

- Non-parametric graphical modelling in dynamical systems is relatively unexplored. Existing approaches rely on specific model assumptions (e.g. linearity, additivity) to establish the consistency of structure recovery even though flexible model families, such as neural networks, are increasingly used in related problems such as feature selection and graphical modelling with static data. In addition, consistent derivative approximations are typically required for consistency arguments which in practice involve choices on the smoothness of the interpolated curve and makes two-step strategies far from automatic and applicable in general dynamical systems.

**Contributions.** This paper establishes the consistency of score-based recovery of $\mathcal{G}$ when an analytic deep neural network model is imposed for $\mathbf{f}$ (such as feed-forward networks with multiple hidden layers and convolutional neural networks) under general observation patterns including irregular sampling. In particular, we consider penalized optimization problems of the form,

$$\arg\min_{\mathbf{f}_\theta} \frac{1}{n} \sum_{i=1}^{n} ||\mathbf{x}(t_i) - \hat{\mathbf{x}}(t_i)||_2^2, \quad \text{subject to} \quad \rho_{n,T}(\mathbf{f}_\theta) \leq \eta \quad \text{and} \quad d\hat{\mathbf{x}}(t) = \mathbf{f}_\theta(\hat{\mathbf{x}}(t))dt, \quad (2)$$

where the observation process $(\mathbf{x}(t_1), \ldots, \mathbf{x}(t_n))$ is given by an irregular sequence of time points $0 \le t_1 < \cdots < t_n \le T$. $\rho_{n,T}(\mathbf{f}_\theta)$ is an adaptive group lasso constraint on the parameter space of $\mathbf{f}_\theta$. We analyze this problem with fixed dimension $d$ and increasing sample size $n$ and horizon $T$ – the sample complexity of this problem depending both on the frequency of sampling $n$ as well as on the time horizon $T$.

A second contribution is to propose an instantiation of this method using differential equations with vector fields parameterized by neural networks (Chen et al., 2018) to model the mean process of (1) with the advantage of implicitly inferring variable derivatives instead of involving a separate approximation step (that is common in the dynamical systems literature). This construction shows that, empirically, graphical models in continuous-time can be inferred accurately in a large range of settings despite irregularly-sampled multivariate time series data and non-linear underlying dependencies.

Code associated with this work may be found at `https://github.com/alexisbellot` and at `https://github.com/vanderschaarlab/mlforhealthlabpub`.

## 2 RELATED WORK

A substantial amount of work devoted to graphical modelling has considered the analysis of penalized least squares and its variants, most prominently in the high-dimensional regression literature with $i.i.d$ data, see e.g. (Friedman et al., 2008; Zou, 2006; Zhao & Yu, 2006). Closely related to our results are a number of extensions that have considered parameter identification in neural networks, using for instance a sparse one-to-one linear layers (Li et al., 2016), group lasso constraints of the input layer of parameters (Zhang et al., 2019) and input to output residual connections (Lemhadri et al., 2021). For a large class of neural networks Dinh & Ho (2020) proved the consistency of adaptive regularization methods. The distinction with our formalism in (2) is that the observations are not corrupted by $i.i.d.$ noise (since successive samples are correlated) and therefore standard concentration inequalities are not sufficient.

Learning graphical models with dependent noise terms is also a topic of significant literature in the context of Granger causality, proposed by Granger (1969) and also popularized by Sims (1980) within autoregressive models. Various authors have considered the consistency of penalized vector autoregression models and proposed tests of Granger causality using parameter estimates in these models, see e.g. (Nardi & Rinaldo, 2011; Kock & Callot, 2015; Adamek et al., 2020; Chernozhukov et al., 2019), and extended some of these approaches to models of neural networks, see e.g. (Tank et al., 2018; Khanna & Tan, 2019; Marcinkevičs & Vogt, 2021) (without however proving consistency of inference). Methods exist also using conditional independence tests such as those given by Runge et al. (2017) and transfer entropy principles originating in Schreiber (2000). The conceptual and statistical contrasts between discrete and continuous accounts of the underlying structural model are substantial and are discussed in the Appendix A.

In the context of differential equations, penalized regression has been explored using two-stage collocation methods, first proposed by Varah (1982), by which derivatives are estimated on smoothed data and subsequently regressed on observed samples for inference. The consistency of parameter estimates has been established for linear models in parameters, as done for example in (Ramsay et al., 2007; Chen et al., 2017; Wu et al., 2014; Brunton et al., 2016). From a modelling perspective, our approach in contrast is end-to-end, coupling the estimation of the underlying paths $\mathbf{x}$ and the vector field $\mathbf{f}$. Graphical modelling has also been considered for linear stochastic differential equations by Bento et al. (2010). Similarly to the discrete-time literature, proposals exist for recovering non-linear vector fields via neural networks (see e.g. (Raissi et al., 2017; Bellot & van der Schaar, 2021)) and Gaussian processes (see e.g. (Heinonen et al., 2018; Wenk et al., 2020)) but we are not aware of any identifiability guarantees.

## 3 GRAPHICAL MODELLING IN CONTINUOUS-TIME

We consider the underlying structure of an evolving process to be described by a multivariate dynamical system of $d$ distinct stochastic processes $\mathbf{x} = (x_1, \ldots, x_d) : [0, T] \to \mathcal{X}^d$ with each instantiation in time $x_j(t)$ for $j = 1, \ldots, d$ and $t > 0$ defined in a bounded open set $\mathcal{X} \subset \mathbb{R}$.

**Definition 1** (Neural Dynamic Structural Model (NDSM)). *We say that $\mathbf{x} = (x_1, \ldots, x_d) : [0, T] \rightarrow \mathcal{X}^d$ follows a Neural Dynamic Structural Model if there exist functions $f_1, \ldots, f_d \in \mathcal{F}$ such that $f_j : \mathcal{X}^d \rightarrow \mathbb{R}$ and,*

$$dx_j(t) = f_j(\mathbf{x}(t))dt + dw_j(t), \qquad \mathbf{x}(t_0) = \mathbf{x}_0, \qquad t \in [0, T], \tag{3}$$

*with $\mathcal{F}$ defined as the space of analytic feed-forward neural networks with sets of parameters $\theta \in \Theta$ defined in bounded, real-valued intervals and $w_j(t)$ is standard Brownian motion independently generated across processes $j$[1].*

We will write $\mathbf{f}_{\theta_0} = (f_1, \ldots, f_d)$ for the true underlying vector field, parameterized by a set of parameter values $\theta_0$. It will be useful to define each layer of each network precisely. Let $A_1^j \in \mathbb{R}^{d \times h}$ denote the $d \times h$ weight matrix (we omit biases for clarity) in the input layer of $f_j, j = 1, \ldots, d$. Let $A_m^j \in \mathbb{R}^{h \times h}$, for $m = 2, \ldots, M - 1$, denote the weight matrix of each hidden layer, and let $A_M^j \in \mathbb{R}^{h \times 1}$ be the $h \times 1$ dimensional output layers of each sub-network such that,

$$f_j(\mathbf{X}) := \phi(\cdots \phi(\phi(\mathbf{X}A_1^j)A_2^j) \cdots) A_M^j, \qquad j = 1, \ldots, d, \tag{4}$$

where $\phi(\cdot)$ is an analytic activation function (e.g. tanh, sigmoid, arctan, softplus, etc.) and $\mathbf{X} \in \mathbb{R}^{n \times d}$ is the sequence of $n$ $d$-dimensional instantiations of $\mathbf{x}$.

**Assumption 1** (Observation process). *The data in practice, is a partial sequence of observations of $\mathbf{x}$ at $n$ time points $(t_1, \ldots, t_n)$ sampled from a temporal point process with positive intensity such that,*

$$(\mathbf{x}_1, \ldots, \mathbf{x}_n) \sim \mathcal{N}(\mu, \Sigma_n), \tag{5}$$

*with a dependency structure encoded in $\Sigma_n \in \mathbb{R}^{n \times n}$. The closer in time two observations are the more closely correlated we can expect them to be. We assume the data to be normalized, i.e. diagonal elements of $\Sigma_n$ to be equal to 1. $\mu$ are the instantiations of the mean process that can be described by an system of ordinary differential equations $d\mathbf{x}(t) = \mathbf{f}(\mathbf{x}(t))dt, \mathbf{x}(0) = \mathbf{x}_0, t \in [0, T]$.*

Time points at which observations are made are thus themselves assumed stochastic, driven by an independent temporal point process with intensity $\lim_{dt \rightarrow 0} Pr(\text{Observation in } [t, t + dt] | \mathcal{H}_t) > 0$ for any $t > 0$ with respect to a filtration $\mathcal{H}_t$ that denotes sigma algebras generated by any sequence of prior observations. Perfectly homogeneous and systematic subsampling has measure zero under this probability model. This is important because it will enable, in principle, to infer local conditional independencies (defined below) arbitrarily well with increasing sample size.

## 3.1 GRAPHICAL PRESENTATION

The stochastic process $\mathbf{x}$ by itself defines a local independence model that can be used to characterize asymmetric dependencies within stochastic processes (Eichler & Didelez, 2012; Eichler, 2013; Didelez, 2012).

**Definition 2** (Local independence). *A process $x$ is locally independent of $y$ given $z$ if, for each time point $t$, the past up until time $t$ of $z$ gives us the same predictable information about $\mathbb{E}(x_t | \mathcal{H}_t(y, z))$ as the past of $x$ and $y$ until time $t$, where $\mathcal{H}_t(y, z)$ is the filtration generated by $y$ and $z$ up to time $t$.*

This independence structure may be represented by a (cyclic) directed graph $\mathcal{G} = (\mathbf{V}, \mathbf{E})$, where each process is associated with a distinct vertex in $\mathbf{V}$ and there is a directed edge $(x_k \rightarrow x_j) \in \mathbf{E}$ if and only if there exist no conditioning subset of $\mathbf{V}$ such $x_k \in \mathbf{V}$ is locally conditionally independent of $x_j \in \mathbf{V}$.

**Lemma 1** (Uniqueness of local independence graphs, Proposition 3.6 (Mogensen et al., 2020)). *In the context of Neural Dynamic Structural models, two processes are locally dependent given any subset of other processes if and only if $x_k$ appears in the differential equation of $x_j$, i.e. $||\partial_k f_j||_{L_2} \neq 0$. Moreover, for any $\mathbf{f}'$ such that $||\partial_k f_j'||_{L_2} = 0$ there exists an equivalent vector field $\mathbf{f}$ such that the euclidian norm of its column vectors $||[A_1^j]_{\cdot k}||_2 = 0$.*

*Proof.* All proofs are given in Appendix B.

---

[1] For analytic function spaces $\mathcal{F}$, the vector field is locally Lipschitz, i.e., the system is stable and the diffusion process has a unique stationary measure that is Gaussian. We assume unique solutions also as $T \rightarrow \infty$.

This Lemma specifies an equivalence relation between functional dependence graphs given by the underlying dynamical system and local independence graphs[2]. It is clear that enforcing $||[A_1^j]._k||_2 = 0$ will remove the local dependence of the $j$-th stochastic process on the $k$-th stochastic process but it is not the case that all functions $\mathbf{f}$ with this particular local independence ($||\partial_k f_j||_{L_2} = 0$) necessarily have zero-valued $k$-th column in its parameters $A_1^j$. This proposition shows that in such cases there exists an equivalent vector field (i.e. that defines the exact same input-output map) that does have $||[A_1^j]._k||_2 = 0$. Local independence graphs may be recovered by searching for such a solution, in theory, if the stochastic process is fully observed.

In practice, with finite samples and complex functions, the map between model and data may not necessarily be identifiable, and a priori should not be expected for highly parameterized neural networks (e.g. a simple rearrangement of the nodes in the same hidden layer leads to a new configuration that produces the same mapping as the generating network). We define next local consistency as a desirable and target property for estimators in practice.

**Definition 3** (Local consistency). *An estimator* $\mathbf{f}_\theta = (f_1, \ldots, f_d)$ *is locally consistent if for any* $\delta > 0$*, there exists* $N_\delta$ *and* $T_\delta$ *such that for* $n > N_\delta$ *and* $T > T_\delta$*, we have* $||\partial_k f_j||_{L_2} \neq 0$[3]*, for all* $k, j \in \{1, \ldots, d\}$ *such that* $x_k$ *is locally significant for* $x_j$*, and have* $||\partial_k f_j||_{L_2} = 0$ *otherwise, with probability at least* $1 - \delta$*.*

### 3.2 FINITE-SAMPLE IDENTIFIABILITY

Write $\mathcal{R}_n(\mathbf{f}_\theta) = \frac{1}{n} \sum_{i=1}^n ||\mathbf{x}(t_i) - \hat{\mathbf{x}}(t_i))||_2^2$ (the dependence on $\mathbf{f}_\theta$ is implicit in $\hat{\mathbf{x}}$) and $\mathcal{R}(\mathbf{f}_\theta)$ for its population counterpart. The difficulty arises from the geometry of the loss function around the set of loss minimizers,

$$\Theta^\star = \{\theta \in \Theta : \mathcal{R}(\mathbf{f}_\theta) = \mathcal{R}(\mathbf{f}_{\theta_0})\}, \tag{6}$$

where $\Theta$ is the parameter space. The set $\Theta^\star$ (of all weight vectors that produce the same input-output map as the generating model) can be quite complex and the behavior of a generic estimator in this set not necessarily reflect local dependencies.

It is possible, however, to constrain the solution space to a subset of "well-behaved" optima for which $G$ is uniquely identifiable even if the full set of parameters $\theta$ is not. It is sufficient for identifiablity of the local independence model to recover the group structure of input layer parameters $[A_1^j]$ exactly. In the context of analytic vector fields $\mathbf{f}$, we define the adjacency matrix $G \in \{0, 1\}^{d \times d}$ associated with $\mathcal{G}$ such that $G_{kj} \neq 0$ if and only if $||\partial_k f_j||_{L_2} \neq 0$. If this pattern can be recovered exactly, then the inferred $G$ corresponds to the functional structure. Optimization with this group structure desiderata has been often considered before as a penalized optimization problem,

$$\underset{\mathbf{f}_\theta}{\arg\min} \quad \mathcal{R}_n(\mathbf{f}_\theta), \quad \text{subject to} \quad d\mathbf{x}(t) = \mathbf{f}_\theta(\mathbf{x}(t))dt \quad \text{and} \quad \rho(\mathbf{f}_\theta) \leq \eta, \tag{7}$$

where we have suppressed the dependence of $\rho$ on the sample size and time horizon for readability. Two popular constraints are the group lasso (GL) and the adaptive group lasso (AGL), see e.g. (Zou, 2006; Zhao & Yu, 2006), defined as,

$$\rho_{\text{GL}}(\mathbf{f}_\theta) := \lambda_{\text{GL}} \sum_{k,j=1}^d ||[A_1^j]._k||_2, \qquad \rho_{\text{AGL}}(\mathbf{f}_\theta) := \lambda_{\text{AGL}} \sum_{k,j=1}^d \frac{1}{||[\hat{A}_1^j]._k||_2^\gamma} ||[A_1^j]._k||_2,$$

respectively, where $\hat{A}_1^j$ is the GL estimate to problem (7), $\lambda_{\text{GL}}, \lambda_{\text{AGL}}$ determine the regularization strength and may vary with $n$ and $T$, $\gamma > 0$ and $|| \cdot ||_2$ is the Euclidian norm. As with other adaptive lasso estimators, AGL uses its base estimator to provide a rough data-dependent estimate to shrink groups of parameters with different regularization strengths. As $n$ and $T$ grow, the weights for non-significant features get inflated while the weights for significant ones remain bounded, allowing AGL to exactly identify significant parameters.

---

[2]Note, however, that this is not true in general for a marginalization of the local independence graph (that we do not consider), i.e. in the context of unobserved processes (Mogensen et al., 2020), in which case several graphs may encode the same set of local independence relations and only equivalence classes may be identifiable.

[3]$\partial_k f_j$ denotes the partial derivative with respect the $k$-th argument of $f_j$, $|| \cdot ||_{L_2}$ is the functional $L_2$ norm.

The following generalization bound will be useful to define convergence rates for penalized solutions.

**Lemma 2** (Generalization bound). *Assume $\Sigma_n$ to be invertible and let $\alpha = (\alpha_1, \ldots, \alpha_n)$ such that $\alpha_1 > \cdots > \alpha_n > 0$ are its eigenvalues. For any $\delta > 0$, there exists $C_\delta > 0$ such that,*

$$|\mathcal{R}_n(f_\theta) - \mathcal{R}(f_\theta)| \leq C_\delta \left( \frac{||\alpha||_2}{n} \right) \sqrt{\log \left( \frac{n}{||\alpha||_2} \right)}, \tag{8}$$

*with probability at least $1 - \delta$.*

**Remark on interpretation.** Note that $||\alpha||_2 \leq ||\alpha||_1 = n$ (the data is assumed to be scaled to have variance 1 for all observations) and that larger values of $||\alpha||_2$ occur with a greater difference in magnitude in the entries of $\alpha$. The difference in magnitude in the principal components of $\mathbf{X}$ is defined by the dependence between samples. A strong dependence between samples (as would be expected with frequently observed time series) leads to proportionally larger magnitude of $\alpha_1$ (the first component of $\alpha$) as more variance in the data is explained by a single direction of variation and thus decreases the effective sample size – making $||\alpha||_2$ closer to $n$. The bound formalizes the trade-off between the number of samples and their dependence. By increasing the observation frequency one can produce an arbitrarily large number of distinct samples but samples become more dependent and therefore less useful for concentration of the empirical error around its population value (unless one simultaneously increases the time horizon $T$).

We now show convergence and local consistency of the adaptive group lasso penalized estimator. The following lemmas use a similar proof technique to Dinh & Ho (2020) with the difference that the convergence speeds differ due to sample dependency.

**Lemma 3** (Convergence of Adaptive Group Lasso). *Let $\tilde{\theta}_n \in \Theta$ be the parameter solution of (7) with adaptive group lasso constraint. For any $\delta > 0$, assuming that $\lambda_{AGL} \to 0$ there exists $v > 0, C_\delta > 0, N_\delta > 0$ and $T_\delta > 0$ such that,*

$$\min_{\theta \in \Theta^*} ||\tilde{\theta}_n - \theta|| \leq C_\delta \left( \lambda_{AGL} + \left( \frac{||\alpha||_2}{n} \right) \sqrt{\log \left( \frac{n}{||\alpha||_2} \right)} \right)^{\frac{1}{\nu}}, \tag{9}$$

*with probability at least $1 - \delta$.*

**Lemma 4** (Local consistency of Adaptive Group Lasso). *Let $\gamma > 0, \epsilon > 0, \nu > 0, \lambda_{AGL} = \Omega((\frac{n}{||\alpha||_2})^{-\gamma/\nu+\epsilon})$, and $\lambda_{AGL} = \Omega(\lambda_{GL}^{\gamma+\epsilon})$, then the adaptive group lasso (solution to problem (7)) is locally consistent.*

**Remark on interpretation.** There exists a well defined time complexity , i.e., a minimum time interval such that, observing the system at an appropriate frequency enables us to reconstruct the network with high probability. The sample complexity is inversely proportional to the time spacing between samples. Lemma 3 implies that with increasing sample size and time horizon the graph $G$ defined such that $[G]_{jk} = 0$ if and only if $||[\tilde{A}_1^j]_{\cdot k}||_2 = 0$ is the local independence graph with high probability. The group lasso will generally not be locally consistent because it forces all parameters to be equally penalized. Some evidence for this claim in the context of feature selection was provided by Zou (2006).

Lemma 4 gives an *asymptotic* guarantee on structure learning. With finite samples, for the estimated structure to have good accuracy for $G$, we have to require that the local dependencies between processes (that define the non-zero entries of $G$) are "sufficiently large" for a given effective sample size. We make a minimum restricted strength assumption whose form reads as,

$$|A_1|_{\min} > C_\delta \left( \lambda_{AGL} + \left( \frac{||\alpha||_2}{n} \right) \sqrt{\log \left( \frac{n}{||\alpha||_2} \right)} \right)^{\frac{1}{\nu}}, \tag{10}$$

where we have defined $|A_1|_{\min} := \min\{||[A_1^j]_{\cdot k}||_2 : j, k = 1, \ldots, d, \quad ||\partial_k f_j||_{L_2} \neq 0\}$ to be the minimum column norm of first layer parameters among all locally dependent stochastic processes. This condition on the design of the problem is not testable but versions of it are essentially necessary in the context of structure learning via parameter estimation (Van de Geer et al., 2011; Zhao & Yu, 2006), and it allows us to describe next a guarantee on the finite sample consistency of structure learning with the Adaptive Group Lasso.

**Lemma 5** (Finite sample local consistency of Adaptive Group Lasso). *Under the conditions of Lemma 4 with the additional minimum restricted strength assumption in (10) on the problem design for particular values of $n$ and $\alpha$, the Adaptive Group Lasso recovers the structure $G$ exactly with high probability.*

### 3.3 ALGORITHM: NEURAL GRAPHICAL MODELLING

Neural ODEs proposed by Chen et al. (2018) are a family of continuous-time models which can be used to define the mean process of a stochastic differential equation explicitly to be the solution to an ODE initial-value problem in which $\mathbf{f}_\theta$ is a free parameter specified by a neural network. For each estimate of $\mathbf{f}_\theta$, the forward trajectory can be computed using any numerical ODE solver:

$$\hat{\mathbf{x}}(t_1), \ldots, \hat{\mathbf{x}}(t_n) = \text{ODESolve}(\mathbf{f}_\theta, \mathbf{x}(t_0), t_1, \ldots, t_n), \tag{11}$$

and thus implicitly enforcing the constraint $d\mathbf{x}(t) = \mathbf{f}_\theta(\mathbf{x}(t))dt$ while optimizing for $\mathcal{R}_n(\mathbf{f}_\theta)$ and thus solving for (2). Gradients with respect to $\theta$ may be computed with adjoint sensitivities and a gradient descent algorithm can be used to backpropagate through the ODE solver and the continuous state dynamics to update the parameters of $\mathbf{f}_\theta$, as shown by Chen et al. (2018).

**Remark on optimization.** The adaptive group lasso constraint is not differentiable and non-separable which precludes applying coordinate optimization algorithms. However, a wide variety of techniques from optimization theory have been developed to tackle this case. A general way of doing so is through proximal optimization, see e.g. (Parikh & Boyd, 2014) that leads to exact zeros in the columns of the input matrices without having to use a cut-off value for selection. The proximal step for the group lasso penalty is given by a group soft-thresholding operation on the input weights and can be interleaved with conventional gradient update steps. Please find all details in the Appendix.

We call this algorithm for structure learning and graphical modelling the Neural Graphical Model (**NGM**).

## 4 EXPERIMENTS

This section makes performance comparisons on controlled experiments designed to analyzed 4 important challenges for graphical modelling with time series data: the **irregularity** of observation times, the **sparsity** of observation times, the **non-linearity** of dynamics, and the **differing scale** of processes in a system.

We benchmark NGM against a variety of algorithms, namely: Three representative vector autoregression models: Neural Granger causality (Tank et al., 2018) in two instantiations, one based on feed forward neural networks (**NGC-MLP**) and one based on recurrent neural networks (**NGC-LSTM**), and the Structural Vector Autoregression Model (**SVAM**, (Hyvärinen et al., 2010), an extension of the LiNGAM algorithm to time series). A representative independence-based approach to structure learning with time series data: **PCMCI** (Runge et al., 2017), extending the PC algorithm. A representative two-stage collocation method we call Dynamic Causal Modelling (**DCM**) in which derivatives are first estimated on interpolations of the data and a penalized neural network is learned to infer $G$ (extending the linear models of (Ramsay et al., 2007; Wu et al., 2014; Brunton et al., 2016)).

**Metric.** We seek to recover the adjacency matrix of local dependencies $G$ between the state of all variables and their variation. All experiments are repeated 100 times and we report mean and standard deviations of the false discovery rate (FDR) and true positive rate (TPR) in recovery performance of $G$. Thresholds for determining the presence and absence of edges in $G$ were chosen for maximum $F_1$ score. For applications in biology, false positives and false negatives can have very different failure interpretations; we choose to report both TPR and FDR explicitly to emphasize the trade-offs of each method when used in practice. Comparisons based on the area under the ROC curve (evaluating the whole range of possible thresholds), experiments comparing different regularization schemes, hyperparameter configurations, dimensionality of processes and run times, as well as details regarding neural network architectures and implementation software may be found in Appendix C.

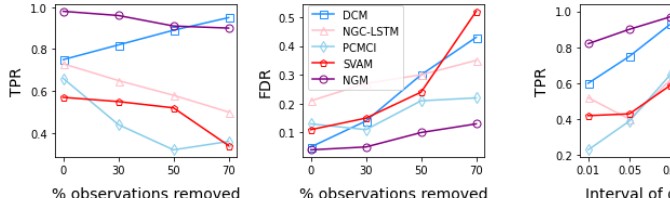
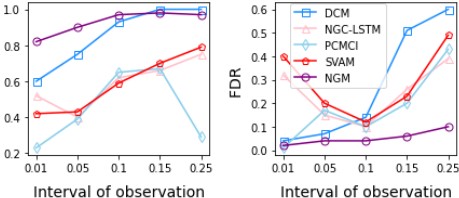

(a) Performance with irregular observation times.      (b) Performance with varying interval of observation.

Figure 2: True positive (higher better) and false discovery (lower better) performance comparisons on Lorenz's model. Thresholds are chosen for maximum $F_1$ score. We omitted plotting NGC-MLP which gave very similar results to NGC-LSTM. NGM is the proposed approach.

### 4.1 LORENZ'S CHAOTIC MODEL

We begin by considering **irregularly** sampled data and **sparsely** sampled data to investigate the benefit of modelling dynamics continuously in time.

We use Lorenz's model (Lorenz, 1996) as an example of the kind of chaotic systems observed in biology e.g., electrodynamics of cardiac tissue (Goldberger & West, 1987), gene regulatory networks (Heltberg et al., 2019), etc. The continuous dynamics in a $d$-dimensional Lorenz model are,

$$\frac{d}{dt}x_i(t) = (x_{i+1}(t) - x_{i-2}(t)) \cdot x_{i-1}(t) - x_i(t) + F + \sigma dw_i(t), \qquad i = 1, \ldots, d$$

where $x_{-1}(t) := x_{d-1}(t)$, $x_0(t) := x_p(t)$, $x_{d+1}(t) := x_1(t)$, $F$ is a forcing constant which determines the level of non-linearity and chaos in the series and $w_i(t)$ is standard independent Brownian motion across $i = 1, \ldots, d$. The initial state of each variable is sampled from a standard Gaussian distribution, $d$ is set to 10, $F$ to 10 and $\sigma$ to 0.5. We illustrate sample trajectories of this system in Figure 1.

- Irregularly sampled data is generated by removing randomly a percentage of the regularly sampled data (with a 0.1 time interval between observations and 1000 observations). For consistency, discrete-time methods here use cubic spline interpolations evaluated at regular time intervals.
- Frequently and sparsely sampled data is generated by varying the time interval of observation (the data always being regularly sampled).

**Results.** Performance results are given in Figure 2. It is important here two look at both TPR and FDR panels for each experiment. For instance, a model returning always a fully connected graph $G$ will have TPR= 1 but FDR= 0. Thus looking at both measures together, NGM significantly improves performance over competing approaches. And moreover, NGM's performance is more robust (worsens less) with increasingly irregular sampling and with increasingly sparse data. The behaviour of discrete-time methods is highly heterogeneous. NGC-LSTM, SVAM and PCMCI are highly dependent on the interval of observation, both too frequent and too sparse measurement times leading to poor FDR. Similarly, as we introduce more irregular sampling TPR decreases and FDR increases which we hypothesize is due to error introduced in the interpolation. This pattern is consistent with our intuition (illustrated in Figure 1) that direct and indirect effects become indistinguishable in discrete time and thus have high FDR with irregular or sparse data. In the case of DCM, it has good performance with frequently observed time series and regular data but rapidly deteriorates otherwise which we hypothesize is due to worsening approximations of derivatives in those cases.

### 4.2 RÖSSLER'S HYPERCHAOTIC MODEL

Next, we consider data with **non-linear** dynamics, said to exhibit *hyperchaotic* behaviour, to demonstrate the flexibility of learned functional relationships.

Chaotic systems, such as Lorenz's model, are characterized by one direction of exponential spreading. If the number of directions of spreading is greater than one the behavior of the system is hyperchaotic

|  | **Rössler** ($d = 10$) | | **Rössler** ($d = 50$) | | **Glycolysis** | |
|---|---|---|---|---|---|---|
|  | **TPR** ↑ | **FDR** ↓ | **TPR** ↑ | **FDR** ↓ | **TPR** ↑ | **FDR** ↓ |
| NGC-MLP | .45 (.05) | .55 (.06) | .31 (.04) | .67 (.06) | .60 (.04) | .51 (.05) |
| NGC-LSTM | .49 (.04) | .53 (.04) | .38 (.04) | .64 (.08) | .69 (.04) | .40 (.04) |
| SVAM | .17 (.03) | .84 (.04) | .03 (.08) | .95 (.09) | .61 (.02) | .53 (.07) |
| PCMCI | .10 (.03) | .92 (.02) | .09 (.06) | .89 (.06) | .56 (.05) | .43 (.06) |
| DCM | .87 (.01) | .10 (.04) | .97 (.01) | .31 (.07) | .67 (.04) | .49 (.05) |
| **NGM (ours)** | .96 (.01) | .02 (.01) | .95 (.01) | .04 (.01) | .84 (.04) | .44 (.09) |

Table 1: Performance comparisons on Rössler's model and the yeast Glycolysis model.

(see e.g. (Barrio et al., 2015)), and much more complicated to predict. In practice this has been observed in chemical reactions (Eiswirth et al., 1992) and EEG models of the brain (Dafilis et al., 2013). Rossler (1979) the first hyperchaotic system of differential equations and here we consider a generalization of this model to arbitrary dimensions and non-linear vector fields as in (Meyer et al., 1997). The $d-$dimensional generalized Rössler model is given by,

$$\frac{d}{dt}x_1(t) = ax_1(t) - x_2(t) + \sigma dw_1(t),$$
$$\frac{d}{dt}x_i(t) = \sin(x_{i-1}(t)) - \sin(x_{i+2}(t)) + \sigma dw_i(t), \quad i = 2, \ldots, d-1,$$
$$\frac{d}{dt}x_d(t) = \epsilon + bx_d(t) \cdot (x_{d-1}(t) - q) + \sigma dw_d(t).$$

We use typical parameters for hyperchaotic behaviour: $a = 0, \epsilon = 0.1, b = 4, q = 2$, as in (Meyer et al., 1997). This system is observed over a sequence of 1000 time points with a 0.1 time unit interval after randomly initializing each variable to a sample from a standard Gaussian distribution and $d$ is set to 10 and 50 to evaluate also performance in higher dimensions. $\sigma$ is set to 0.1.

**Results.** Performance results are given in Table 1 and the contrast between non-linear and linear methods is stark. NGM continues to strongly outperform other methods with almost perfect recovery of the graph $G$ in both low and high-dimensional regimes. The strongest baseline is DCM which also models the underlying graph in continuous-time.

### 4.3    MODEL OF OSCILLATIONS IN YEAST GLYCOLYSIS

We conclude with an experiment from the pharmacology literature which makes extensive use of dynamical models to determine the interaction patterns of drugs in the body. In these systems, it is the **difference in scale** of different biochemicals that makes structure recovery difficult.

The glycolytic oscillator model is a standard benchmark for this kind of systems. It simulates the cycles of the metabolic pathway that breaks down glucose in cells. We simulate the system presented in equation (19) by Daniels & Nemenman (2015) defined by 7 biochemical components and fully described in Appendix C.5.

For biology in particular, under our assumptions, a feature of NGM is that it explicitly discovers **f** and a consistent graphical structure such that the model can be used to simulate the expected effect of interventions by modifying the weights of a trained network which may have a large impact on the design of (laboratory) experiments. We show, for illustration, NGM's simulation of the mean behaviour of the glycolytic oscillator in Figure 3, which successfully recovers the true mean dynamics.

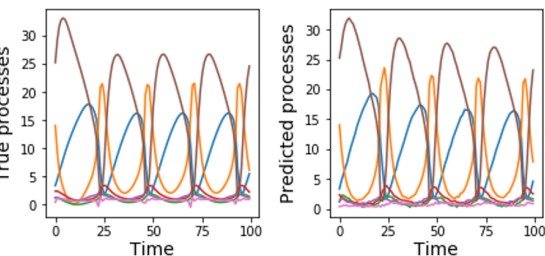

Figure 3: True and NGM-estimated Glycolytic oscillations.

**Results.** Table 1 shows that performance on this data is more heterogeneous. Although NGM outperforms or is competitive with other methods, FDR is high. Better understanding how to model

structure in systems with different scales (see this explicitly in Figure 3 with numerous variables with values equal to a small fraction of the largest observations) thus remains an important challenge.

## 5 DISCUSSION

An emerging trend in biology, climate science and healthcare is the measurement of increasing amounts of datatypes (individual gene transcript levels, protein abundances, molecular concentrations of pollution in the air, biomarkers in the hospital, etc.) on an increasing resolution but with heterogeneous observation patterns. A graphical model that is stable over a large number of variables, over a large number of sampling frequencies and observation patterns has attractive properties to scientists in all these domains. In this paper, we have discussed graphical modelling from a continuous-time perspective and have shown the consistency of penalized least squares problems in general models of differential equations with analytic vector fields. As an instantiation of this method, we propose a novel graphical modelling algorithm, Neural Graphical Model, that models the latent vector field explicitly with penalized extensions to Neural ODEs and is applicable to general irregularly-sampled multivariate time series. We conclude with some additional remarks.

- **Marginalized local independence graphs.** In general, we cannot expect that local independence graphs uniquely identify the underlying functional dependency structure if unobserved processes influence the dynamics of the system. In such cases one can define an equivalence class of local independence graphs which have been characterized recently by Mogensen et al. (2020). We have not considered this setting. Anecdotally however, a form of violation of this assumption was included in the Lorenz system (perturbed by a constant factor $F$ which is not modelled) for which NGM achieves almost perfect structure recovery. Some robustness to violations of the assumption of no unobserved processes may thus be expected although we did not perform a formal investigation of this phenomenon.

- **Switching dynamical systems.** In the context of a Neural ODE, $\mathbf{f}$ was defined using a continuous neural network and $\mathbf{x}(t)$ is always continuous in $t$. Trajectories modeled by an ODE can thus have limited representation capabilities when discontinuities in the state occur. Examples are switching dynamical systems which are hybrid discrete and continuous systems that choose which dynamics to follow based on a discrete switch, and are popular in neuroscience and finance (see e.g. Friston (2009)). For these systems, a practical extension would be to follow Chen et al. (2020) and modify the gradient update to include discrete changes in the event state in which case, to infer the causal graph, partial derivatives $\partial_k f_j$ would be defined piece-wise.

## ACKNOWLEDGEMENTS

This work was supported by the Alan Turing Institute under the EPSRC grant EP/N510129/1, the ONR and the NSF grants number 1462245 and number 1533983.

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

# Appendix

This appendix is outlined as follows:

## A    RELATED WORK

### A.1    GRAPHICAL MODELLING IN DISCRETE-TIME

Graphical modelling with time series has been driven by applications in causality. A first practical definition of causality for inference with time series data was given by Granger (Granger, 1969). A time series $x_i$ is said to *Granger-cause* $x_j$ if omitting the past of a time series $x_i$ in a time series model including $x_j$'s own and other covariates' past increases the prediction error of the next time step of $x_j$. Implementations of this principle are variants of vector auto-regressive (VAR) models and its extensions (see e.g. (Sims, 1980; Chen et al., 2004; Tank et al., 2018; Pamfil et al., 2020)), for example in the linear case assuming,

$$\mathbf{x}(t + \Delta t) = B_1 \mathbf{x}(t) + \cdots + B_k \mathbf{x}(t - k\Delta t) + \epsilon(t), \qquad \epsilon(t) \sim P, \tag{12}$$

for some distribution $P$. Each one of the matrices $B_1, \ldots, B_k$ then describe *lagged* causal relationships with different lags. Subsampling occurs when the frequency of observation is lower than $\Delta t$ and renders VAR models generally unidentifiable although specific exceptions exist and have been explored in (Gong et al., 2015; Danks & Plis, 2013). Alternatives to VAR models include the PC algorithm with conditional independence testing methods accounting for auto-correlations between successive observations as done by Runge (2018); Runge et al. (2017; 2019) and transfer entropy principles (Schreiber, 2000).

We illustrate next with an example why, from a learning perspective, graphical modelling in discrete-time cannot be consistently applied for the purpose of graphical modelling in dynamical systems without strong assumptions on the observation process or parameterization of the underlying structural model.

### A.1.1    EXAMPLE

Assume the true dynamics of two processes $\mathbf{x}(t) = (x_1(t), x_2(t))^\mathsf{T}$ to be given by,

$$\frac{d}{dt}\mathbf{x}(t) = A\mathbf{x}(t), \qquad A = \begin{pmatrix} -1 & 2 \\ -4 & -0.5 \end{pmatrix}, \tag{13}$$

The underlying dynamics are given by $A$: $x_2$ causing an increase in the rate of change of $x_1$ (with value 2) while $x_2$ having a large negative causal effect on the rate of change of $x_1$ (with value $-4$).

The corresponding discrete-time model (with one time lag for simplicity) may be defined as,

$$\mathbf{x}(t + \Delta t) = B_{\Delta t} \cdot \mathbf{x}(t), \tag{14}$$

where $\Delta t$ is the time interval of observation. The two models are connected by a simple relation, given by $B_{\Delta t} = \exp\{A \cdot \Delta t\}$, that can be used to uniquely compute the off-diagonal entries $[B_{\Delta t}]_{12}$ and $[B_{\Delta t}]_{21}$ indicating the strength of "causal" effects $x_2 \to x_1$ and $x_1 \to x_2$ respectively, under the Granger-causality paradigm (Granger, 1969).

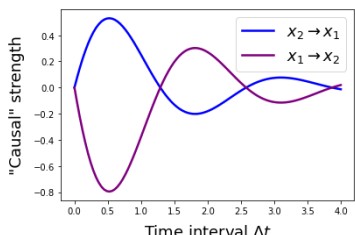

Figure 4: Discrete-time model inferred causal interaction with underlying continuous-time process.

As a first observation, irrespective of the time interval $\Delta t$ note that a single discrete-time model $B_{\Delta t}$ may describe multiple underlying mechanisms $A$ as matrix logarithms $\log B_{\Delta t}$ are not identified if $B_{\Delta t}$ has complex eigenvalues (see e.g. (Yue et al., 2016) for a formal result). Second, for a single model defined by $A$ the discrete-time causal interpretation (i.e. $B_{\Delta t}$) may change dramatically as a function of the time interval $\Delta t$ at which the process is observed, as shown in Figure 4. Describing causality within dynamical systems in discrete-time is inherently an ill-posed problem. In fact, only for dynamical systems that are bivariate, stable, and non-oscillating can we expect consistent conclusions at all measurement intervals.

**Proposition 1** (Causal inconsistency in discrete time (Kuiper & Ryan, 2018)). *The sign of off-diagonal entries of $B_{\Delta t}$ and $A$ agree for all time intervals $\Delta t > 0$ if $A$ defines a bivariate, stable, and non-oscillating system of differential equations.*

*Proof.* (Given here for completeness – can also be found in (Kuiper & Ryan, 2018)) Using the relationship $B_{\Delta t} = \exp\{A \cdot \Delta t\}$ we can write,

$$B_{\Delta t} = V^{-1} \exp\{D_{A \cdot \Delta t}\} V \tag{15}$$

where $D_{A \cdot \Delta t}$ represents the diagonal matrix containing the scalar exponentials of the eigenvalues of $A$ multiplied by the scalar $\Delta t$ and $V$ is the matrix of eigenvectors. This can be used to relate two estimated discrete-time matrix coefficients with each other,

$$B_{\Delta_2 t} = B_{\Delta_1 t}^{(\Delta_2 t / \Delta_1 t)} \tag{16}$$

Consider now a bivariate process with eigenvalues of an estimated $B_{\Delta_1 t}$ denoted by $\lambda_1$ and $\lambda 2$, and let $\Delta_2 t = n \cdot \Delta_1 t$. Then, misleading causal conclusions are obtained if the sign of the entries of $B_{\Delta_2 t}$ differ from $B_{\Delta_1 t}$. In this case they may be computed explicitly. It holds that the sign of off-diagonal entries are equal if and only if $(\lambda_1 - \lambda_2)$ (determining the sign of entries of $B_{\Delta_1 t}$) and $(\lambda_1^n - \lambda_2^n)$ (determining the sign of entries of $B_{\Delta_2 t}$) are equal. This is the case if both $\lambda_1$ and $\lambda_2$ lie between $0$ and $1$ i.e., a stable, bi-variate, non-oscillating system, but will not hold in general otherwise.

Counter-examples of discrepancies can be found for many other types of differential equations in (Kuiper & Ryan, 2018).

### A.2 RELATED WORK ON MODELLING DIFFERENTIAL EQUATIONS

Two-stage collocation methods were first proposed by Varah (1982). The authors proposed to fit a smoothing estimate $\hat{\mathbf{x}}(\cdot; h, \theta)$ to the observations $y_1, \ldots, y_n$ with a smoothing parameter $h$, and using it and its derivative with respect to $t$ in order to estimate the vector field $\mathbf{f}$ in a system of differential equations,

$$\min_{\theta} \ \frac{1}{n} \sum_{i=1}^{n} ||d\hat{\mathbf{x}}(t_i; h) - \mathbf{f}_\theta(\hat{\mathbf{x}}(t_i; h)))||_2^2, \tag{17}$$

where $\hat{\mathbf{x}}(\cdot; h) = \underset{\mathbf{x}(\cdot; h) \in \mathcal{H}}{\arg\min} \ \frac{1}{n} \sum_{i=1}^{n} ||\mathbf{y}_i - \mathbf{x}(t_i; h)||_2^2$ is the smooth interpolation function and most approaches assume $\mathbf{f}$ to be linear although variations of this principle have been developed using non-linear functions of observations (e.g. see (Dattner et al., 2015)) and using bases of functions without explicitly determining their form (e.g. see (Chen et al., 2017)). There are two important differences between the proposed approach and two-stage collocation methods.

- The fact that one must choose the interpolation and smoothing function leads to a very different analysis of the properties of estimators. Estimates rely heavily on the smoothing estimates obtained consistency has only been shown for certain values of the smoothing parameter $h$ that are hard to choose in practice (Chen et al., 2017).

- The objectives of two-stage collocation methods is the statistical estimation of the parameters $\theta$ of $\mathbf{f}_\theta$ rather than of the parameters of the graphical structure induced by $\mathbf{f}_\theta$. As a contrast, in the particular case of analytic neural networks the exact set of parameters in the data generating mechanism is not uniquely identified as multiple alternatives define the same input-output map of the vector field $\mathbf{f}_\theta$, even in the infinite-sample regime.

**Broader literature.** Modelling time series is a wide and varied research topic. One can train feed-forward or recurrent neural networks to approximate a differential equation (Chen et al., 2018; Raissi, 2018) and model interventions (Bellot & van der Schaar, 2021). Gaussian Processes have been adapted to fit differential equations (Raissi et al., 2017) and have been proposed to model continuous-time interventions (Schulam & Saria, 2017; Soleimani et al., 2017). And tree-based methods are also popular in biology to model irregularly sampled data (Huynh-Thu & Sanguinetti, 2015; Bellot & Schaar, 2020). The objective in these papers however is to extrapolate the latent state of the process i.e., the forward problem: inferring the latent state of the trajectory from time series. This description has not yet involved structure learning which, in contrast, is concerned with the inverse problem i.e., using discrete measurements of trajectories to infer the underlying structural model.

### A.3 Philosophical debate about the nature of causality

A mechanistic interpretation of causality views causal claims as claims about the existence of a mechanism or process that mediates events at one time with events at a later time and has been formalized mathematically as a system of structural differential equations. Proponents of this view hold that the metaphysical connection between mechanisms and causality is very close: two events are causally connected if and only if they are connected by an underlying physical mechanism. Glennan (Glennan, 2010; 2002) and Machamer (Machamer, 2004) define a mechanism as a *complex system* of interacting parts, but *process* accounts have also been proposed (Salmon, 1984; Dowe, 1992) that focus on the fact that causal processes manifest a conserved quantity. In the latter, an interaction between two processes is causal if there is an exchange of a conserved quantity between them. Mechanistic theories of causality, from a philosophical standpoint, are normally contrasted with probabilistic (Pearl, 2009; Spirtes et al., 2000), counterfactual (Rubin, 2005; Lewis, 1979; 1998) and manipulationist (Pearl, 2009; Price, 1991; Woodward, 1984) theories of causality. These define a connection to be causal if and only if a change in one makes a difference to the other. This distinction is a useful conceptual contrast even though it has been noted that these accounts overlap in some measure: for instance, mechanisms can be given a counterfactual analysis and thus would be a form of difference-making theory (Williamson, 2011).

Differential equations and their causal semantics (e.g. formalizing the meaning of interventions or counterfactuals) are often derived from those of (static or time-independent) Structural Causal Models (SCMs, Definition 7.1.1 (Pearl, 2009)) that, in dynamical systems, have been interpreted as defining an intermediate layer of expressiveness between the underlying model of differential equations and statistical models based on associations (Schölkopf, 2019). The connection between these conceptual layers is of ongoing interest. For instance, SCMs have been shown to describe changes in equilibrium states (if they arise) (Dash, 2005; Mooij et al., 2013), and have been shown to describe changes in asymptotic dynamics (Rubenstein et al., 2016) under carefully defined interventions, and has been the focus of most work at the intersection of dynamical systems and causality.

In any case, the underlying nature and semantics of causality does not influence the theory and algorithms presented in this paper as long as we assume that the causal system of interest may be represented as a Neural Dynamic Structural model. In such a model, causation across time in dynamical systems is due to a derivative (e.g., velocity $d\mathbf{x}$) causing a change in its integral (e.g., position $\mathbf{x}$). All other causation is contemporaneous, occurring between two variables on a time-scale that is smaller than the time-step of observation and defined by the sparsity pattern of $\mathbf{f}$.

# B PROOFS

We will use $\mathcal{R}_n(\mathbf{f}_\theta)$ and $\mathcal{R}_n(\theta)$ interchangeably. With the notation introduced in the main body of this paper, we will use the following Lemma that extends the standard Taylor's inequality around a local optimum to unbounded sets $\theta^*$.

**Lemma 7** (Lemma 3.2. (Dinh & Ho, 2020)) There exist $c_2, \nu > 0$ and such that $|\mathcal{R}(\theta) - \mathcal{R}(\theta_0)| \geq c_2 d(\theta, \Theta^*)^\nu$ for all $\theta \in \Theta$.

$d(\theta, \Theta^*)$ is the minimum Euclidian norm between $\theta$ and any element of $\Theta^*$.

## B.1 PROOF OF LEMMA 1

The uniqueness of local independence graphs in systems of fully observed stochastic processes was given in Proposition 3.6 by Mogensen et al. (2020). We restate the Lemma for convenience.

**Lemma 1** (Uniqueness of local independence graphs, Proposition 3.6 (Mogensen et al., 2020)). *In the context of Neural Dynamic Structural models, two processes are locally dependent given any subset of other processes if and only if $x_k$ appears in the differential equation of $x_j$, i.e. $||\partial_k f_j||_{L_2} \neq 0$. Moreover, for any $\mathbf{f}'$ such that $||\partial_k f'_j||_{L_2} = 0$ there exists an equivalent vector field $\mathbf{f}$ such that the euclidian norm of its column vectors $||[A_1^j]_{\cdot k}||_2 = 0$.*

*Proof.* Consider the class of NDSMs with $\mathcal{F}$ such that $f_j$ is independent of process $\mathbf{x}_k$, that is $||\partial_k f_j||_{L_2} = 0$, and the class of NDSMs $\mathcal{F}_0$ with $f_j$ parameterized such that $||[A_1^j]_{\cdot k}||_2 = 0$. We will how that $\mathcal{F} = \mathcal{F}_0$.

It is clear that the class of NDSMs with $\mathcal{F}_0$ is contained in the class of NDSMs with $\mathcal{F}$ as a process $\mathbf{x}_k$ interacts with the vector field $f_j$ of $\mathbf{x}_j$ only if the corresponding entries in the first layer of the analytic neural network are non-zero: $\mathcal{F}_0 \subset \mathcal{F}$.

For the converse consider a NSDM whose vector field $f_j$ of $\mathbf{x}_j$ is independent of the $k$-th process $\mathbf{x}_k$ and consider a set of processes $\tilde{\mathbf{x}}$ and $\mathbf{x}$ such that $\tilde{\mathbf{x}} = \mathbf{x}$ except for the $k$-th entry of $\tilde{\mathbf{x}}$ which is set to the zero function $\tilde{\mathbf{x}}_k : [0, T] \to 0$. Because of independence, and because the two processes differ only in their $k$-th entry: $f_j(\tilde{\mathbf{x}}(t)) = f_j(\mathbf{x}(t))$.

Now define $\tilde{A}_1$ be the matrix such that $[\tilde{A}_1]_{ik} = 0$ and $[\tilde{A}_1]_{ik'} = [A_1]ik$ for all $k' \neq k$. Then it holds that $\tilde{A}_1 \mathbf{x}(t) = A_1 \tilde{\mathbf{x}}(t)$. And therefore also $f_j(\mathbf{x}(t); A_1) = f_j(\tilde{\mathbf{x}}(t); A_1) = f_j(\mathbf{x}(t); \tilde{A}_1)$. By definition of $\mathcal{F}_0$, $f_j(\cdot; \tilde{A}_1) \in \mathbf{F}^0$ and thus $\mathcal{F} \subset \mathcal{F}_0$. ∎

## B.2 PROOF LEMMA 2

We restate the Lemma for convenience.

**Lemma 2** (Generalization bound). *Assume $\Sigma_n$ to be invertible and let $\alpha = (\alpha_1, \ldots, \alpha_n)$ such that $\alpha_1 > \cdots > \alpha_n > 0$ be its eigenvalues. For any $\delta > 0$, $\frac{n}{||\alpha||_2} > 3$, there exists a $C > 0$ such that,*

$$|\mathcal{R}_n(\mathbf{f}_\theta) - \mathcal{R}(\mathbf{f}_\theta)| \leq \left(\frac{||\alpha||_2}{n}\right) \sqrt{C \log\left(\frac{n}{||\alpha||_2}\right)}, \tag{18}$$

*with probability at least $1 - \delta$.*

*Proof.* Without loss of generality, we consider each differential equation separately. $n\mathcal{R}_n(\mathbf{f}_\theta) =: X^T X$ is a sum of dependent squared normal random variables. With this notation $X = (X_1, \ldots, X_n) \in \mathbb{R}^n$ where $X_i = (Y_{ij} - \hat{x}_j(t_i)) \in \mathbb{R}$ is a univariate random variable and $Y_{ij} \in \mathbb{R}$ is the $j$-th random variable at time $t_i$ given of the observation model and underlying dynamical system. $X = (X_1, \ldots, X_n)$ has a joint distribution defined by a mean $\mu$ and covariance matrix $\Sigma_n$, as defined in the main body of this paper. We may write $Z = \Sigma_n^{-1/2}(X - \mu)$ and,

$$X^T X = (Z + \Sigma_n^{-1/2}\mu)\Sigma_n(Z + \Sigma_n^{-1/2}\mu). \tag{19}$$

Let $\Sigma_n = V^T A V$ be the eigendecomposition of $\Sigma_n$ where $V$ is an orthogonal basis of eigenvectors and $A$ is a diagonal matrix of positive eigenvalues $\alpha = (\alpha_1, \ldots, \alpha_n)$. $U = VZ$ then is also

multivariate normal, with expectation zero and identity covariance matrix (since $V^T V = V V^T = I_n$). For $u = V \Sigma_n^{-1/2} \mu$, rewriting the decomposition above in terms of $U$ and $u$

$$X^T X = (U + u)^T A (U + u) = \sum_{i=1}^{n} \alpha_i (U_i + u_i)^2. \tag{20}$$

The distribution of $n \mathcal{R}_n(\mathbf{f}_\theta) = X^T X$ is thus a weighted non-central $\chi^2$ random variable.

Write $W = V \Sigma_n^{-1/2} \in \mathbb{R}^{n \times n}$. By applying the concentration results in Theorem 6 and 7 in (Zhang & Zhou, 2020) we have,

$$\mathbb{P}(|\mathcal{R}_n(\theta) - \mathcal{R}(\theta)| > s) \le \exp \left\{ \frac{-C_1 n^2 s^2}{||\alpha||_2^2 + 2 \sum_{i=1}^{n} \left( \sum_{j=1}^{n} W_{ij}(f(x(t_j); \theta) - f(x(t_j); \theta_0)) \right)^2} \right\} \tag{21}$$

$$\le \exp \left\{ -C_2 \left( \frac{ns}{||\alpha||_2} \right)^2 \right\}, \tag{22}$$

for all $s$ such that,

$$0 < s < \frac{||\alpha||_2}{n ||\alpha||_\infty} + \frac{2 \sum_{i=1}^{n} \left( \sum_{j=1}^{n} W_{ij}(f(x(t_j); \theta) - f(x(t_j); \theta_0)) \right)^2}{n}. \tag{23}$$

$C_1, C_2 > 0$ are two scalars not depending on $n$ or $\alpha$. The remaining of the proof follows the argument of (Dinh & Ho, 2020). We define events,

$$\mathcal{A}(\theta, s) = \{ |\mathcal{R}_n(\theta) - \mathcal{R}(\theta)| > s \}, \tag{24}$$

$$\mathcal{B}(\theta, s) = \{ \exists \theta' \in \Theta \text{ such that } ||\theta' - \theta||_2 \le \frac{s}{4 M_\delta} \text{ and } |\mathcal{R}_n(\theta') - \mathcal{R}(\theta')| > s \} \tag{25}$$

$$\mathcal{C} = \{ |\mathcal{R}_n(\theta) - \mathcal{R}_n(\theta')| \le M_\delta ||\theta - \theta'||_2, \forall \theta, \theta' \in \Theta \}, \tag{26}$$

where the last event $\mathcal{C}$ is defined with respect to the Lipschitz constant $M_\delta$ of $\mathcal{R}_n$ (assumed to be Lipschitz with probability at least $1 - \delta$, that is $\mathbb{P}(\mathcal{C}) \ge 1 - \delta$). Let $m = \dim(\Theta)$, there exist $C_3(m) \ge 1$ and a finite set $\mathcal{H} \subset \Theta$ such that,

$$\Theta \subset \bigcup_{\theta \in \mathcal{H}} \mathcal{V}(\theta, \epsilon), \qquad |\mathcal{H}| \le C_3/\epsilon^m, \tag{27}$$

where we choose $\epsilon = s/(4 M_\delta)$. $\mathcal{V}(\theta, \epsilon)$ denotes the open ball centered at $\theta$ with radius $\epsilon$, and $|\mathcal{H}|$ denotes the cardinality of $\mathcal{H}$. In other words, $\mathcal{H}$ $\epsilon$-covers $\Theta$ and the inequality involving the cardinality of $\mathcal{H}$ follows because $\Theta$ is a bounded subset of Euclidian space, see e.g. section 27.1 in (Shalev-Shwartz & Ben-David, 2014). By a union bound over all elements in $\mathcal{H}$,

$$\mathbb{P}\left( \exists \theta \in \mathcal{H} : |\mathcal{R}_n(\theta) - \mathcal{R}(\theta)| > s \right) \le C_3 (4 M_\delta)^m s^{-m} \exp \left\{ -C_2 \left( \frac{n}{||\alpha||_2} \right)^2 s^2 \right\}. \tag{28}$$

Since $\mathcal{B}(\theta, s) \cap \mathcal{C} \subset \mathcal{A}(\theta, s)$ and $\mathcal{H} \subset \Theta$ we have,

$$\mathbb{P}(\exists \theta \in \Theta : |\mathcal{R}_n(\theta) - \mathcal{R}(\theta)| > s) \le C_3 (4 M_\delta)^m s^{-m} \exp \left\{ -C_2 \left( \frac{n}{||\alpha||_2} \right)^2 s^2 \right\} + \delta. \tag{29}$$

Now let $k = \frac{n}{||\alpha||_2}$ and let $s = \frac{\sqrt{C \log(k)}}{k}$ for notational simplicity. To complete the proof we need to choose $C$ such that,

$$C_3 \left( \frac{4 M_\delta k}{\sqrt{C \log(k)}} \right)^m \exp \left\{ -C_2 \cdot C \cdot \log(k) \right\} \le \delta. \tag{30}$$

This inequality holds if,

$$C_3(4M_\delta)^m \cdot k^{m-C_2 \cdot C} \leq \delta, \tag{31}$$

since $(C \log(k))^{m/2} \geq 1$, which can be obtained if $m - C_2 \cdot C > 0$ and $C_3(4M_\delta)^m \cdot 3^{m-C_2 \cdot C} \leq \delta$, since $k > 3$ by assumption, so that,

$$C \geq \frac{1}{C_2 \log(3)} \left( \log(C_3) + m \log(4M_\delta) + \log(1/\delta) \right). \tag{32}$$

∎

### B.3 PROOF LEMMA 3

To traverse this result, we will start by considering the convergence of the group lasso.

**Lemma 7** (Convergence of Group Lasso). *For any $\delta > 0$, assuming that $\lambda_{GL} \to 0$ there exists $v > 0, C_\delta > 0, N_\delta > 0$ and $T_\delta > 0$ such that,*

$$\min_{\theta \in \Theta^*} ||\hat{\theta}_n - \theta|| \leq C_\delta \left( \lambda_{GL}^{\nu/\nu-1} + \left( \frac{||\alpha||_2}{n} \right) \sqrt{\log \left( \frac{n}{||\alpha||_2} \right)} \right)^{\frac{1}{\nu}}, \tag{33}$$

*with probability at least $1 - \delta$.*

*Proof.* Recall the group lasso and adaptive group lasso penalty terms,

$$\rho_{GL}(\theta) := \lambda_{GL} \sum_{k,j=1}^d ||[A_1^j]_{\cdot k}|| \qquad \text{and} \qquad \rho_{AGL}(\theta) := \lambda_{AGL} \sum_{k,j=1}^d \frac{1}{||[\hat{A}_1^j]_{\cdot k}||^\gamma} ||[A_1^j]_{\cdot k}||.$$

By definition, we have,

$$\mathcal{R}_n(\hat{\theta}_n) + \rho_{GL}(\hat{\theta}_n) \leq \mathcal{R}_n(\theta_0) + \rho_{GL}(\theta_0), \tag{34}$$

where $\hat{\theta}_n = \arg \min_{\theta \in \Theta} \mathcal{R}_n(\theta) + \rho_{GL}(\theta)$ is the parameter solution to the group lasso.

It holds then that,

$$\min_{\theta \in \Theta^*} c_2 ||\hat{\theta}_n - \theta||_2^\nu \leq \mathcal{R}(\hat{\theta}_n) - \mathcal{R}(\theta_0) \tag{35}$$

$$\leq |\mathcal{R}(\hat{\theta}_n) - \mathcal{R}_n(\hat{\theta}_n)| + |\mathcal{R}(\theta_0) - \mathcal{R}_n(\theta_0)| + |\mathcal{R}_n(\hat{\theta}_n) - \mathcal{R}_n(\theta_0)| \tag{36}$$

$$\leq 2 \left( \frac{||\alpha||_2}{n} \right) \sqrt{C \log \left( \frac{n}{||\alpha||_2} \right)} + |\rho_{GL}(\theta_0) - \rho_{GL}(\hat{\theta}_n)| \tag{37}$$

$$\leq 2 \left( \frac{||\alpha||_2}{n} \right) \sqrt{C \log \left( \frac{n}{||\alpha||_2} \right)} + \lambda_{GL} \cdot K \cdot ||\theta_0 - \hat{\theta}_n||_2, \tag{38}$$

where the first inequality is due to Lemma 6 (for some $c_2, \nu > 0$), the second inequality is due to the triangle inequality, the third inequality is due to equation (34) and Lemma 1, and the fourth inequality comes from the Lipschitzness of $\rho_{GL}$. We have used $K > 0$ to denote the Lipschitz constant of $\rho_{GL}$. The last step is given by Youngs's inequality, e.g. as stated in section 5.1 (Dinh & Ho, 2020), to conclude that,

$$\min_{\theta \in \Theta^*} ||\hat{\theta}_n - \theta||_2^\nu \leq C_\delta \left( \left( \frac{||\alpha||_2}{n} \right) \sqrt{\log \left( \frac{n}{||\alpha||_2} \right)} + \lambda_{GL}^{\frac{\nu}{\nu-1}} \right). \tag{39}$$

for some constant $C_\delta$. ∎

We will now state and prove Lemma 3.

**Lemma 3** (Convergence of Adaptive Group Lasso). *For any $\delta > 0$, assuming that $\lambda_{AGL} \to 0$ there exists $v > 0, C_\delta > 0, N_\delta > 0$ and $T_\delta > 0$ such that,*

$$\min_{\theta \in \Theta^*} ||\tilde{\theta}_n - \theta|| \leq C_\delta \left( \lambda_{AGL} + \left( \frac{||\alpha||_2}{n} \right) \sqrt{\log \left( \frac{n}{||\alpha||_2} \right)} \right)^{\frac{1}{\nu}}, \tag{40}$$

*with probability at least* $1 - \delta$.

*Proof.* By the convergence of the group lasso $||[\hat{A}_1^j]_{\cdot k}||_2$ is bounded away from zero for any process $k$ that causally significant for process $j$, $k, j \in \{1, \ldots, d\}$. Let the set of causally significant pairs $(k, j)$ be denoted $\mathcal{S}$. Then we can define,

$$\mathcal{M}(\theta) = \sum_{(k,j) \in S} \frac{1}{||[\hat{A}_1^j]_{\cdot k}||_2^\gamma} ||[A_1^j]_{\cdot k}||_2 < \infty. \tag{41}$$

since $||[\hat{A}_1^j]_{\cdot k}||_2 > 0$.

Let $\tilde{\theta}_n = \underset{\theta \in \Theta}{\arg\min} \ \mathcal{R}_n(\theta) + \rho_{\mathrm{AGL}}(\theta)$ be the solution parameters of the adaptive group lasso problem. By a similar derivation to that used in (35),

$$\mathcal{R}(\tilde{\theta}_n) - \mathcal{R}(\theta_0) \le 2C\left(\frac{||\alpha||_2}{n}\right)\sqrt{\log\left(\frac{n}{||\alpha||_2}\right)} + \lambda_{\mathrm{AGL}} \cdot (\mathcal{M}(\tilde{\theta}_n) - \mathcal{M}(\theta_0)) \tag{42}$$

$$\le 2C\left(\frac{||\alpha||_2}{n}\right)\sqrt{\log\left(\frac{n}{||\alpha||_2}\right)} + \lambda_{\mathrm{AGL}}\mathcal{M}(\tilde{\theta}_n). \tag{43}$$

And since $\mathcal{M}(\tilde{\theta})$ is a finite positive scalar, again by a similar derivation to that used in (35),

$$\min_{\theta \in \Theta^*} \ ||\tilde{\theta}_n - \theta||_2 \le C_\delta \left(\left(\frac{||\alpha||_2}{n}\right)\sqrt{\log\left(\frac{n}{||\alpha||_2}\right)} + \lambda_{\mathrm{AGL}}\right)^{1/\nu}. \tag{44}$$

for some constant $C_\delta > 0$. $\blacksquare$

### B.4 Proof of Lemma 4

We restate the Lemma for convenience.

**Lemma 4** (Local consistency of Adaptive Group Lasso). *Let* $\gamma > 0$, $\epsilon > 0$, $\nu > 0$, $\lambda_{AGL} = \Omega((\frac{n}{||\alpha||_2})^{-\gamma/\nu+\epsilon})$, *and* $\lambda_{AGL} = \Omega(\lambda_{GL}^{\gamma+\epsilon})$, *then the adaptive group lasso is locally consistent.*

*Proof.* By the convergence of the Group Lasso, for any pair $(j, k)$ of non-significant processes,

$$||[\hat{A}_1^j]_{\cdot k}||_2 \le C_\delta\left(\lambda_{\mathrm{GL}}^{\nu/\nu-1} + \left(\frac{||\alpha||_2}{n}\right)\sqrt{\log\left(\frac{n}{||\alpha||_2}\right)}\right)^{\frac{1}{\nu}}, \tag{45}$$

with probability at least $1 - \delta$. It holds therefore that,

$$\lim_{\frac{n}{||\alpha||_2} \to \infty} \frac{1}{||[\hat{A}_1^j]_{\cdot k}||_2^\gamma} \ge \infty. \tag{46}$$

Now assume for contradiction that there exists a pair $(k, j)$ of non-locally significant processes (that is, $x_k$ is not locally significant for $x_j$) such that $||[\tilde{A}_1^j]_{\cdot k}||_2 \ne 0$ (the $\sim$ notation above the matrix $A_1^j$ denotes estimation with the adaptive group lasso) and define $\phi(\tilde{\theta}_n)$ to be equal to $\tilde{\theta}_n$ except that non-significant parameters are set to zero. By the definition of $\tilde{\theta}_n$ as minimizing the empirical risk regularized by the adaptive group lasso constraint,

$$\mathcal{R}_n(\tilde{\theta}_n) + \lambda_{\mathrm{AGL}}\frac{1}{||[\hat{A}_1^j]_{\cdot k}||_2^\gamma}||[\tilde{A}_1^j]_{\cdot k}||_2 \le \mathcal{R}_n(\phi(\tilde{\theta}_n)). \tag{47}$$

Since the regularization term on the right-hand side is zero by the definition of $\phi(\tilde{\theta})$. Then by Assumption 3,

$$\lambda_{\mathrm{AGL}}\frac{1}{||[\hat{A}_1^j]_{\cdot k}||_2^\gamma}||[\tilde{A}_1^j]_{\cdot k}||_2 \le \mathcal{R}_n(\phi(\tilde{\theta}_n)) - \mathcal{R}_n(\tilde{\theta}_n) \tag{48}$$

$$\le M_\delta||\phi(\tilde{\theta}_n) - \tilde{\theta}_n||_2 \tag{49}$$

$$= M_\delta||[\tilde{A}_1^j]_{\cdot k}||_2. \tag{50}$$

But since we have assumed $||[\tilde{A}_1^j]._k||_2 \neq 0$ it follows from above that $\lambda_{\text{AGL}} \frac{1}{||[\hat{A}_1^j]._k||_2^\gamma} \leq M_\delta$ which is a contradiction of Lemma 7 that proved the convergence of the group lasso and in particular that $\lim_{\frac{n}{||\alpha||_2} \to \infty} \frac{1}{||[\hat{A}_1^j]._k||_2^\gamma} = \infty$ for non-locally significant pairs of processes $(k, j)$. ∎

## B.5 PROOF OF LEMMA 5

We restate the Lemma for convenience.

**Lemma 5** (Finite sample local consistency of Adaptive Group Lasso). *Under the conditions of Lemma 4 with the additional minimum restricted strength assumption in (10) on the problem design for particular values of $n$ and $\alpha$, the Adaptive Group Lasso recovers the structure $G$ exactly with high probability.*

*Proof.* First, note that for the set of loss minimizers $\Theta^\star$ defined in eq. (6) and by using the fact that neural networks are analytic, it does hold that for any two locally dependent processes $x_k$ and $x_j$ the first layer parameters of any model with minimum loss are bounded away from zero, i.e. $||[A_1^j]._k||_2 \geq c$ for some $c > 0$.

To see this, assume for a contradiction that no such $c$ exists, and therefore that there exists $[\tilde{A}_1^j]._k \in \Theta^\star$ such that $||[A_1^j]._k||_2 = 0$ since neural networks are analytic and each one of the parameters is defined in bounded intervals. This would imply that there exists a neural network with the same input-output relationship as the true model $f_j$ that does not depend on its $k$-th input, which is a contradiction because $||\partial_k f_j||_{L_2} \neq 0$.

Next, given the minimum strength condition on the column norms of first layer parameters related to locally dependent processes,

$$|A_1|_{\min} > C_\delta \left( \lambda_{\text{AGL}} + \left( \frac{||\alpha||_2}{n} \right) \sqrt{\log \left( \frac{n}{||\alpha||_2} \right)} \right)^{\frac{1}{\nu}} \tag{51}$$

where recall that $|A_1|_{\min} := \min\{||[A_1^j]._k||_2 : j, k = 1, \ldots, d, \quad ||\partial_k f_j||_{L_2} \neq 0\}$, we have that by Lemma 3 that estimated parameters $||\tilde{A}_1^j]._k||_2$ are bounded away from zero for specific values of $\alpha$ and $n$ since,

$$\min_{\theta \in \Theta^*} \quad ||[\tilde{A}_1^j]._k - [A_1^j]._k||_2 \leq C_\delta \left( \lambda_{\text{AGL}} + \left( \frac{||\alpha||_2}{n} \right) \sqrt{\log \left( \frac{n}{||\alpha||_2} \right)} \right)^{1/\nu}, \tag{52}$$

with high probability. ∎

## C  EXPERIMENTAL DETAILS

### C.1  RESULTS AS A FUNCTION OF FEATURE DIMENSIONALITY, NEURAL NETWORK PARAMETERIZATION AND RUN TIME COMPARISONS

Performance with increasing number of variables is monitored to some extent with the Rössler experiment. We extent this analysis in this section to include performance comparisons as a function of more variables. We also include run-time comparisons and performance comparisons as a function of different model parameterizations to understand the practical use of the proposed approach. We limit our comparisons here to NGC-LSTM, DCM and NGM.

| | Rössler ($d = 10$) | | Rössler ($d = 50$) | | Rössler ($d = 100$) | | Rössler ($d = 100$) | |
|---|---|---|---|---|---|---|---|---|
| | **TPR** ↑ | **FDR** ↓ | **TPR** ↑ | **FDR** ↓ | **TPR** ↑ | **FDR** ↓ | **TPR** ↑ | **FDR** ↓ |
| NGC-LSTM | .49 (.04) | .53 (.04) | .38 (.04) | .64 (.08) | - | - | - | - |
| DCM | .87 (.01) | .10 (.04) | .97 (.01) | .31 (.07) | .94 (.04) | .35 (.05) | .90 (.04) | .40 (.05) |
| **NGM (ours)** | .96 (.01) | .02 (.01) | .95 (.01) | .04 (.01) | .95 (.04) | .05 (.02) | .89 (.04) | .05 (.02) |

Table 2: Performance comparisons on Rössler's model.

| | Rössler ($d = 10$) | Rössler ($d = 50$) | Rössler ($d = 100$) | Rössler ($d = 200$) |
|---|---|---|---|---|
| NGC-LSTM | 2523 | 5321 | - | - |
| DCM | 12 | 74 | 320 | 991 |
| **NGM (ours)** | 291 | 480 | 923 | 2289 |

Table 3: Learning time in seconds.

| | Rössler ($d = 10$) | | Rössler ($d = 50$) | |
|---|---|---|---|---|
| | **TPR** ↑ | **FDR** ↓ | **TPR** ↑ | **FDR** ↓ |
| (1,10) – Default | .96 (.01) | .02 (.01) | .95 (.01) | .04 (.01) |
| (1,50) | .95 (.01) | .02 (.01) | .95 (.01) | .04 (.01) |
| (2,10) | .96 (.01) | .02 (.01) | .96 (.01) | .03 (.01) |
| (2,50) | .92 (.01) | .03 (.01) | .96 (.01) | .04 (.01) |
| (5,10) | .95 (.01) | .03 (.01) | .95 (.01) | .05 (.01) |
| (5,50) | .94 (.01) | .04 (.01) | .96 (.01) | .06 (.01) |

Table 4: Performance comparisons with different neural network architectures. Notation: (number of hidden layers, number of hidden units in each hidden layer).

Note that the parameter of interest is a norm over the columns of parameter matrices in the first NN layer which we found to be sufficiently coarse not be sensitive to small variations in architecture choices. Our default parameterization worked well across all datasets we analyzed. In Table 4, we show additional experiments that cover 6 possible configurations that span reasonable choices that a practitioner may make. We find that performance results are largely invariant to differences in the number of layers and layer size.

### C.2  RESULTS USING THE AREA UNDER THE ROC CURVE

In this section we report all experiments in the main body of this paper using the area under the ROC curve (AUC) as performance metric. Figure 5 contains comparisons for experiments using the Lorenz model and Table 5 contains comparisons for the Rössler and Glycolytic experiments.

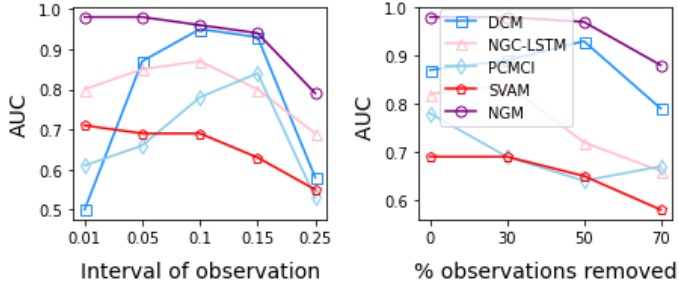

Figure 5: Experiments on Lorenz's model.

|  | **Rössler** ($d = 10$) | **Rössler** ($d = 50$) | **Glycolytic** |
|---|---|---|---|
| NGC-MLP | .73 (.02) | .70 (.02) | .57 (.04) |
| NGC-LSTM | .75 (.03) | .74 (.02) | .65 (.04) |
| SVAM | .56 (.03) | .50 (.05) | .60 (.03) |
| PCMCI | .51 (.04) | .50 (.05) | .62 (.03) |
| DCM | .95 (.01) | .90 (.02) | .70 (.02) |
| **NGM (ours)** | .99 (.01) | .98 (.01) | .78 (.03) |

Table 5: AUC on Rössler and Glycolytic data. Numbers in parenthesis are standard deviations.

## C.3 PURELY SYNTHETIC EXPERIMENT

This experiment investigates the behaviour of different regularization schemes with a purely synthetic data generating mechanism. We generate 1000 observations of 50 variables $\mathbf{X} \in \mathbb{R}^{1000 \times 50}$ using regular evaluations of the process defined by,

$$\frac{d}{dt}x_j(t) = g_j(\mathbf{x}(t)) + dw_j(t), \qquad \mathbf{x}(0) = \mathbf{x}_0, \tag{53}$$

for $j = 1, \ldots, 50$, where each vector field component $g_j : \mathbb{R}^{50} \to \mathbb{R}$ is parameterized by a neural network with three hidden layers of 10 nodes, such that $g_j$ depends only on 5 (locally significant) of its 50 arguments. The initial state $\mathbf{x}_0$ as well as all weights and biases are independently drawn from standard Gaussian random variables before setting the locally not significant first layer columns to zero. The time interval between observations is fixed at 0.1 units.

The results are given in Table 6. We write $\text{NGM}_{GL}$ for NGM with group lasso regularization, $\text{NGM}_{AGL}$ for NGM with adaptive group lasso regularization and $\text{NGM}_L$ for NGM with conventional lasso regularization. True positive rates are comparable across regularization schemes but false discovery rates are much lower for adaptive regularization which suggest that standard lasso and group lasso algorithms may not be aggressive enough to enforce sparsity strictly. As a consequence, we may need cut-off values to interpret $\text{NGM}_{GL}$ locally which are difficult to specify in practice, while $\text{NGM}_{AGL}$ sets most non-local processes to zero exactly without further post-processing.

|  | **TPR** ↑ | **FDR** ↓ |
|---|---|---|
| $\text{NGM}_L$ | .63 (.04) | .30 (.08) |
| $\text{NGM}_{GL}$ | .71 (.05) | .25 (.08) |
| $\text{NGM}_{AGL}$ | .70 (.04) | .17 (.05) |

Table 6: Regularization choices.

### C.4 ALGORITHM IMPLEMENTATION

#### C.4.1 NEURAL GRAPHICAL MODELLING (NGM)

**Proximal gradient descent.** The proximal step for the group lasso penalty is given by a group soft-thresholding operation on the input weights.

In each iteration, proximal-gradient steps make two update computations to the relevant parameters, denoted here $\theta \in \{[A_1^j]_{\cdot k}, j = 1, \ldots, d\}$. They are,

1. $\theta \leftarrow \theta - \alpha \nabla \mathcal{L}(\theta)$

2. $\theta \leftarrow \operatorname{argmin}_{w \in \mathbb{R}^d} ||w - \theta||^2 + \alpha \lambda_{\mathrm{GL},n} \sum_{k,j=1}^{d} ||[A_1^j]_{\cdot k}||$

where the second part is the proximal operator with respect to the constraint, which for each $\theta \in [A_1^j]_{\cdot k}$ is equivalent to a soft-threshold group-wise update,

$$[A_1^j]_{\cdot k} \leftarrow \frac{[A_1^j]_{\cdot k}}{||[A_1^j]_{\cdot k}||} \max\{0, ||[A_1^j]_{\cdot k}|| - \alpha \lambda_{\mathrm{GL},n}\} \tag{54}$$

Regularizing constants are chosen from the set $\{0.001, 0.01, 0.05, 0.1, 0.5, 1, 2\}$ with $\gamma = 2$ using average test errors from random train-test splits of the corresponding dataset.

**Threshold selection.** With this optimization procedure we do not need to set a threshold for converting the weights to the presence / absence of edges in the graph. A non-zero estimate of $||[A_1^j]_{\cdot k}||$ is considered as presence of an edge in the underlying graph and a zero estimate of $||[A_1^j]_{\cdot k}||$ is considered as absence of an edge.

**Architecture.** The integrand $\mathbf{f}_\theta$ was taken to be a feed-forward neural network as described with a single hidden layer of size 10 and elu activation functions after each layer except after the output layer. In each case we used the Adam optimiser as implemented by PyTorch. Starting learning rates varied between experiments (with values between 0.001 and 0.01) before being reduced by half if metrics failed to improve for a certain number of epochs. It was enough in all experiments to consider the final parameter configuration (instead of the one with best validation performance) as only norms of first layer parameters are of interest which we found not to be sensitive to the exact epoch choice. The same architecture was used for all experiments.

#### C.4.2 NEURAL GRANGER CAUSALITY

We implement Neural Granger Causality (Tank et al., 2018) with the code provided by the authors at `https://github.com/iancovert/Neural-GC`.

We considered two architectures: an MLP to fit the lagged time series explicitly to its next value in time, and a LSTM to model the hidden state capturing the relevant history information. We follow the author's implementation and use a single hidden layer with 5 nodes, 5 lagged variables, relu activation function and hierarchical penalty optimized with their GISTA training procedure.

**Threshold selection.** NGC similarly uses an adaptive procedure to optimize parameters and presence / absence of edges in the underlying graph can simply be read off as the non-zero parameters.

#### C.4.3 DYNAMIC CAUSAL MODELLING

Dynamic causal modelling attempts to recover the vector field $\mathbf{f}$ explicitly by fitting a multivariate linear model to map the set of variables $\mathbf{X}(t) \in \mathbb{R}^d$ with estimated derivatives $d\mathbf{X}(t) \in \mathbb{R}^d$ (in our case computed separately with smoothing spline approximations).

**Architecture.** In our implementation, we parameterize $\mathbf{f}$ with neural networks. We use a single hidden layer of size 10 and elu activation functions after each layer except after the output layer. In each case we used the Adam optimiser as implemented by PyTorch.

The variant of Dynamic Causal Modelling we used, with neural networks to approximate the mapping between variable states and their derivatives, is our own implementation. The architecture is similar

to that used in NGM with the exception that derivatives are approximated a priori with the derivatives of natural cubic splines taken to interpolate the observed data. The smoothing hyperparameter in the spline fit was chosen for visual inspection to preserve the trajectory of the curves in each data separately.

The optimization problem thus consisted in fitting the vector field $\mathbf{f}_\theta$ explicitly such as to fit the approximated derivatives $\frac{d\mathbf{x}}{dt}(t)$ evaluated at a given time $t$ as well as possible. Derivatives are computed from a cubic spline interpolation of the time series. We manually tune interpolation hyperparameters to obtain a visually faithful approximation of the observed trajectory for each dataset.

The optimization objective is given as,

$$\arg\min_{\mathbf{f}_\theta \in \mathcal{F}} \quad \frac{1}{n}\sum_{i=1}^{n}\left\|\frac{d\mathbf{X}}{dt}(t_i) - \mathbf{f}_\theta(\mathbf{x}(t_i))\right\|_2^2 + \rho_{\text{AGL}}(\mathbf{f}_\theta). \tag{55}$$

$\mathbf{X}$ here denotes the interpolation over time of the time series. We use similar regularization arguments for consistency with NGM and also proximal gradient descent for optimization.

**Threshold selection.** For DCM, in our implementation, we use the same proximal gradient descent method with the adaptive group lasso constraint and thus we do not require a threshold to determine the presence / absence of edges in the underlying graph. Presence / absence of edges is defined by non-zero parameter values.

### C.4.4 PCMCI

PCMCI (Runge et al., 2017; Runge, 2018; Runge et al., 2019) is a *discrete-time* two-step approach that uses a version of the PC-algorithm with the momentary conditional independence test to account for autocorrelation in the time series.

PCMCI was implemented with the python package provided by the authors at `https://jakobrunge.github.io/tigramite/`.

**Threshold selection.** We chose to adjust for multiple testing with Benjamini-Hochberg's procedure and considered associations significant, determining presence / absence of edges, with $p$-values below 0.00001 (chosen here because it gave a good trade-off between TPR and FDR i.e., the values with maximum $F_1$ score).

### C.4.5 SVAM

SVAM (Hyvärinen et al., 2010) was implemented with the implementation provided by the authors at `https://github.com/cdt15/lingam` with the BIC model selection criterion and a single lagged variable. Including more lagged variables did not alter our results much but adds an additional choice as to how to define causality as we would have multiple estimated matrices of inter-relationships.

**Threshold selection.** We chose the threshold for converting the weights to the presence / absence of edges in the graph based on $F_1$ scores on validation data and was consistent (around 0.1) across datasets.

### C.5 DETAILS ON GLYCOLYSIS EXPERIMENT

Pharmacology is a branch of medicine that makes extensive use of dynamical models to determine the interaction patterns of drugs in the body. Like many other systems, the nonlinearity of dynamics in biology makes it hard to infer drug interactions from experimental data. Simple linear models are computationally efficient, but cannot incorporate these important nonlinearities. The glycolytic oscillator model is a standard benchmark for these kinds of systems. It simulates the cycles of the metabolic pathway that breaks down glucose in cells. We simulate the system presented in Daniels and Nemenman (Daniels & Nemenman, 2015) in their equation 19, mimicking glycolytic oscillations in yeast cells.

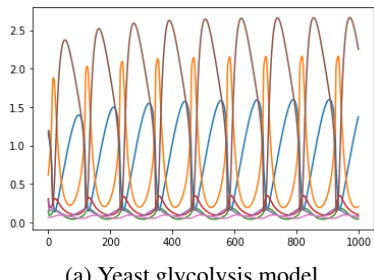

(a) Yeast glycolysis model.

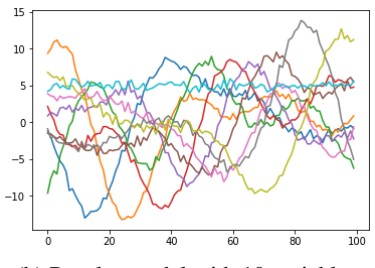

(b) Rössler model with 10 variables.

Figure 6: Sample trajectories.

The dynamics, defined by 7 biochemical components denoted $x_1, x_2, \ldots, x_7$, are given by the following system of equations (Daniels & Nemenman, 2015),

$$\frac{d}{dt}x_1(t) = 2.5 - \frac{100 \cdot x_1(t) \cdot x_6(t)}{1 + (x_6(t)/0.52)^4} + 0.01 dw_1(t)$$

$$\frac{d}{dt}x_2(t) = 2 \cdot \frac{100 \cdot x_1(t) \cdot x_6(t)}{1 + (x_6(t)/0.52)^4} - 6x_2(t) \cdot (1 - x_5(t)) - 12x_2(t)x_5(t) + 0.01 dw_2(t)$$

$$\frac{d}{dt}x_3(t) = 6x_2(t) \cdot (1 - x_5(t)) - 16 \cdot x_3(t) \cdot (4 - x_6(t)) + 0.01 dw_3(t)$$

$$\frac{d}{dt}x_4(t) = 16 \cdot x_3(t) \cdot (4 - x_6(t)) - 100 \cdot x_4(t) \cdot x_5(t) - 13 \cdot (x_4(t) - x_7(t)) + 0.01 dw_4(t)$$

$$\frac{d}{dt}x_5(t) = 6x_2(t) \cdot (1 - x_5(t)) - 100 \cdot x_4(t) \cdot x_5(t) - 12x_2(t)x_5(t) + 0.01 dw_5(t)$$

$$\frac{d}{dt}x_6(t) = -2 \cdot \frac{100 \cdot x_1(t) \cdot x_6(t)}{1 + (x_6(t)/0.52)^4} + 32 \cdot x_3(t) \cdot (4 - x_6(t)) - 1.28 \cdot x_6(t) + 0.01 dw_6(t)$$

$$\frac{d}{dt}x_7(t) = 1.3 \cdot (x_4(t) - x_7(t)) - 1.8 \cdot x_7(t) + 0.01 dw_7(t)$$

A sample of the trajectories is given in Figure 6, where after the first few time units, the system settles down onto a simple limit-cycle behavior.

As in previous examples, the system is observed over a sequence of $T$ time points with a $0.1$ time unit interval after randomly initializing each variable in the ranges provided in Table 1 of (Daniels & Nemenman, 2015). The data is stacked into two matrices for $\mathbf{X} \in \mathbb{R}^{T \times 7}$ (and $d\mathbf{X} \in \mathbb{R}^{T \times 7}$ for methods using approximated derivatives) where each row of $\mathbf{X}$ is a snapshot of the state $x$ in time.

