# OpenReview forum: "Neural graphical modelling in continuous-time: consistency guarantees and algorithms"
_ICLR.cc/2022/Conference — ICLR 2022 Poster_

### Official Review · Reviewer_xAV7 · 2021-11-02

**Correctness:** 4
**Technical Novelty And Significance:** 3
**Empirical Novelty And Significance:** 3
**Recommendation:** 8
**Confidence:** 3

**Main Review:**

The paper is masterfully written: the presentation is rigorous and yet reads effortlessly. I enjoyed reading the paper throughout.

The presented method is straightforward extension of neural ODEs with adaptive group lasso with proximal methods. The main contribution is the theoretical consistency and convergence analyses of such a system. The paper discusses learning bounds in terms of data fit, and in terms of sparsity pattern. The main insight is to show that one only needs to sparsify the first layer weight of the neural ODE.

The proposed method has outstanding performance in the experiments that cover observation irregularity and sparsity, and a biological and chaotic system. The method is compared against lots of competing methods.

Technical comments:
* Lemma 1: If xk appears in differential of xj, then surely one expects to see non-zero jacobian between them, while here it is implied to be zero instead.


**Summary Of The Paper:**

The paper proposes to learn Jacobian-sparse neural network ODEs from irregular trajectories of a dynamical system. The main contribution is the sparsity of the ODE Jacobian, which results in learning of differential covariate causalities. Learning the differential structure is an important real-world problem. The proposed method is elegant, simple and effective, although incremental.


**Summary Of The Review:**

This is a fantastically written paper with an effective model for an important problem with solid theory and outstanding empirical performance.

---

### Official Review · Reviewer_4Lyp · 2021-11-02

**Correctness:** 4
**Technical Novelty And Significance:** 2
**Empirical Novelty And Significance:** 2
**Recommendation:** 5
**Confidence:** 4

**Main Review:**

The proposed method is a straightforward application of the neural ODE and group Lasso method. The novelty is low. In addition, it is unclear how the proposed method can be used to handle heterogeneous dynamic systems, which are often the case of practical systems and might make the proposed method less robust.

**Summary Of The Paper:**

This paper proposes a penalized neural ODE method for learning graphical models for multivariate dynamic systems.

**Summary Of The Review:**

This paper provides a reasonable method for graphical modeling of dynamic systems, although the technical novelty is limited.

---

### Official Review · Reviewer_TPnG · 2021-11-10

**Correctness:** 4
**Technical Novelty And Significance:** 3
**Empirical Novelty And Significance:** 3
**Recommendation:** 5
**Confidence:** 4

**Main Review:**

Strengths:

-- This is a very well written paper that introduces a new framework for graphical modelling. Reading the paper was a real pleasure. Because I was asked to review the paper as an emergency reviewer, I couldn't afford time to check the proofs.

-- The framework is very pleasant and novel and paves the way for other researchers to build on.

Weaknesses:

-- Unfortunately, my enthusiasm for this paper is somewhat tempered by the theoretical results, which do have potential, but also limitations. In traditional graphical modeling, one is interested in the recoverability of the underlying graph. Here, unless I missed something, in Lemma 3, the authors are proving some results concerning the parameters of the model. Why would we be interested in the parameters? It seems like learning the structure would be able to provide practitioners with more useful qualitative information about the relationships between the processes $x_j$.

-- It's also not clear to me why one would be interested in the generalization bound as stated in Lemma 2. Generalization bounds are useful in supervised learning settings, but here one is interested (as I mentioned), the quality of the graph that is recovered from the observations.

-- Even if we accept that the risk and difference between the estimated parameter and the true parameter are important quantities, it is unclear what the results imply. Can the authors comment about tightness of the upper bounds on the generalization error (8) or the consistency gap in (9)? I realize that a single paper cannot contain too many results, but I fail to appreciate how the bounds in (8) and (9) depend on some hardness parameter of the learning task. I am familiar with graphical models, and usually the results depend on the minimum strength $\beta$ in the exponential family parameterization. Here, this dependence seems to be completely absent.

**Summary Of The Paper:**

This paper introduces a brand new graphical modeling framework from the perspective of neural ODEs. Traditionally structure learning involves using sampled data to learn the structure of graphs. This paper, however, looks at the graph structure learning problem from a different viewpoint, using continuous-time dynamics inspired from neural ODEs. Theoretical guarantees on parameter estimation are provided. Some experiments on benchmark time-series datasets are also conducted.

**Summary Of The Review:**

See main review above. The paper does have a lot of potential, but perhaps it is not fully mature at this point.

---

### Decision · Program_Chairs · 2022-01-20

**Decision:**

Accept (Poster)

**Comment:**

This main focus of this paper is graph modeling. Specifically, this paper considers a setting in which data is generated under continuous time dynamics based on neural ODE. Theoretical results regarding parameter estimation are provided. The results are also supported by experiments.

The reviewers appreciate a thorough response to their questions and think that this paper would be of interest to ICLR and ML community. Please address reviewers comments in your final version.